# Pentacyclic Triterpene Profile and Its Biosynthetic Pathway in *Cecropia telenitida* as a Prospective Dietary Supplement

**DOI:** 10.3390/molecules26041064

**Published:** 2021-02-18

**Authors:** Gustavo Gutiérrez, Laura Marcela Valencia, Deisy Giraldo-Dávila, Marianny Y. Combariza, Elkin Galeano, Norman Balcazar, Aram J. Panay, Alejandra Maria Jerez, Guillermo Montoya

**Affiliations:** 1Department of Pharmaceutical Sciences, School of Natural Sciences, Universidad Icesi, Cali 760031, Colombia; ggutierrezg94@gmail.com (G.G.); lauramarcela.valenciatorres@gmail.com (L.M.V.); 2Escuela de Química, Universidad Industrial de Santander, Bucaramanga 680003, Colombia; deisy.giraldo.davila@gmail.com (D.G.-D.); marianny@uis.edu.co (M.Y.C.); 3Productos Naturales Marinos, Departamento de Farmacia, Facultad de Ciencias Farmacéuticas y Alimentarias, Universidad de Antioquia, UdeA, Calle 70 # 52-21, Laboratorio 2-131, Medellín 050010, Colombia; elkin.galeano@udea.edu.co; 4Department of Physiology and Biochemistry, School of Medicine, Universidad de Antioquia, Carrera 51D Nº 62-29, Medellin 050010, Colombia; norman.balcazar@udea.edu.co; 5GENMOL Group, Sede de Investigación Universitaria, Universidad de Antioquia, Calle 62 # 52-59, Medellín 050010, Colombia; 6Independent Researcher, Calle 28 # 86-70 Apt 712, Cali 760031, Colombia; joelpanay@gmail.com; 7Department of Biomedical Sciences, School of Health, Universidad Icesi, Cali 760031, Colombia; amjerez@icesi.edu.co; 8Center for Specialized and Biotechnological Natural Ingredients (CINEB), School of Natural Sciences, Universidad Icesi, Cali 760031, Colombia

**Keywords:** *Cecropia telenitida*, pentacyclic triterpene, type 2 diabetes, dietary supplement

## Abstract

Promising research over the past decades has shown that some types of pentacyclic triterpenes (PTs) are associated with the prevention of type 2 diabetes (T2D), especially those found in foods. The most abundant edible sources of PTs are those belonging to the ursane and oleanane scaffold. The principal finding is that *Cecropia telenitida* contains abundant oleanane and ursane PT types with similar oxygenation patterns to those found in food matrices. We studied the compositional profile of a rich PT fraction (DE16-R) and carried out a viability test over different cell lines. The biosynthetic pathway connected to the isolated PTs in *C. telenitida* offers a specific medicinal benefit related to the modulation of T2D. This current study suggests that this plant can assemble isobaric, positional isomers or epimeric PT. Ursane or oleanane scaffolds with the same oxygenation pattern are always shared by the PTs in *C. telenitida*, as demonstrated by its biosynthetic pathway. Local communities have long used this plant in traditional medicine, and humans have consumed ursane and oleanane PTs in fruits since ancient times, two key points we believe useful in considering the medicinal benefits of *C. telenitida* and explaining how a group of molecules sharing a closely related scaffold can express effectiveness.

## 1. Introduction

Research reports from the past two decades highlight triterpenes’ potential role in the prevention and eventual treatment of type 2 diabetes. For example, consumption of pentacyclic triterpenes (PTs), found in olive oil, has been associated with benefits such as improved endothelial function in healthy adults [1]. There is also a plethora of research describing the potential of PTs in treating prediabetes and diabetes. According to Sheng and Sun, PTs are able to associate with the biochemical machinery in living systems as they are pre-adapted to interact with different cellular networks in animal and human bodies [2]. In the same report, the authors describe 22 molecular targets of glucose and lipid metabolism modulated by PTs such as oleanolic, ursolic, betulinic, and asiatic acids [2]. However, other PTs like corosolic, maslinic, arjunolic, hederagenic, serjanic, and tormentic acids are also reported to decrease glucose levels, increase insulin secretion and insulin sensitivity, or reduce inflammatory adipokines in adipose tissue [3,4,5,6]. In overlooking the properties of PTs, the global health system potentially risks missing a consistent, efficacious, but most importantly, affordable preventative treatment for one of the leading causes of death and disability worldwide.

The PTs mentioned above feature a common structural scaffold—an oleanyl and ursanyl—reported to modulate some human and animal metabolic pathways. For instance, the proven gene expression inhibition of proinflammatory cytokines by serjanic acid is crucial to prevent or ameliorate pancreatic β-cell damage within a diabetic state, as shown in a preclinical murine model [7]. Various experimental models have demonstrated that islet inflammation plays a significant role in the pathogenesis of β-cell failure patients. The NF-κB deactivation positively regulates insulin receptor substrate IRS1/2, encouraging translocation of the glucose transporter type 4 (GLUT4) through the cellular membrane, decreasing insulin resistance, and contributing significantly to a normoglycemic condition [3,8].

Though other tetracyclic triterpene types, such as ginsenosides, have been clinically proven not to improve β-cell function or insulin sensitivity [9], a considerable number of preclinical reports suggest the opposite [10,11]. This might open the debate regarding the ability of penta- and tetracyclic triterpenes to modulate the same biological targets and would depend on their structures and, therefore, on their biosynthetic pathway. Naturally occurring compounds such as glycyrrhizin, glycyrrhizinic acid, and their semisynthetic derivative carbenoxolone are structurally related to PTs produced by *Cecropia telenitida,* but these compounds have a different oxygenation pattern and therefore can induce hypertension by reducing the transcriptions of both 11β-HSD2 and CYP11B2 in the vasculature, leading to lower aldosterone and higher corticosterone production in vessels [12]. It is probable this specific triterpene scaffold decoration provides licorice with a different target modulation as described for immune or inflammatory based diseases [13].

*Cecropia telenitida*, an endemic plant from South America’s Andean region, contains abundant oleanane and ursane PT types [14]. The genus has been widely used as a hypoglycemic and anti-inflammatory remedy in traditional medicine [15]. In a previous report, using in vitro testing, we isolated a fraction from *Cecropia telenitida* active towards the metabolic target 11β-hydroxysteroid dehydrogenase type 1 (11β-HSD1). The active fraction was codified as DE16 [16], and part of our initial work was to describe its metabolic PT profile. Interestingly, in the same study, DE16 subfractions were considerably less active than the whole fraction over the selected metabolic target (11β-HSD1), suggesting a synergistic effect.

We want to further examine the compositional space associated with the pentacyclic triterpene profile of the DE16 fraction and assess its use as a dietary supplement for prediabetic conditions. To this end, we studied the compositional PTs profile of the leader fraction obtained from another individual (DE16-R) through MALDI-TOF, LC–MS and NMR analyses and tested its viability over different cell lines. The challenging task of describing DE16-R’s metabolic profile proved to be an exciting example of “going back to the roots” in natural product chemistry.

## 2. Results

### 2.1. Retrieving a Chemical DE16 Fraction (DE16-R) from Different Samplings

The automated flash chromatography fractionation of powdered *Cecropia telenitida* roots [16] yielded 86 fractions. TLC fingerprinting, using vanillin-sulfuric acid as a derivatizing agent, was carried out to determine the fractions whose chromatographic profile was similar to DE16. Fractions 40 to 54 displayed the highest TLC chemical profile similarity to DE16. Those fractions were pooled, labeled as DE16-R, and analyzed by MALDI-TOF mass spectrometry to compare with the original DE16 fraction (Figure 1).

Figure 1 shows the ion at *m*/*z* 487.343 as the MS spectra’s base peak in both fractions; some low-abundance species are observed in DE16-R in the lower mass range. All the species observed in the MALDI (-) spectra correspond to deprotonated molecules formed by proton abstraction, from acidic molecules present in the extract, by the highly basic DMAN matrix (1,8-bis(dimethylamino)naphthalene) according to the acid–base Brønsted–Lowry theory. The observation of only a few ionic species in the negative MALDI spectra is somewhat puzzling since the fraction was expected to exhibit a more complex profile. Thus, positive ion mode MALDI MS, using DHB as the matrix, was used to interrogate individual subfractions instead of the pooled fraction. Interestingly, the positive mode MALDI MS spectra for the subfractions showed an ion at *m*/*z* 511.339 as the base peak in all spectra. This ion corresponds to the sodium adduct of the same molecular entity observed at *m*/*z* 487.343 in negative ion mode MALDI.

With some doubts surfacing regarding the MALDI-TOF outcomes, it was decided to make use of the separation dimension coupled to mass spectrometry in the fraction analysis. A negative mode by UPLC-APCI-MS analysis was performed with the aim of generating more information about some molecular deprotonated ions at close values to the more abundant, and likely illustrate more clearly the existence of less concentrated or a lesser amount of unfavored ionized triterpenes.

The results in both fractions shared the high abundance of 487 and 471 *m*/*z* ions in a negative mode, as shown in Figure 2. Other less abundant ions, close to a five-minute retention time in both samples, shared the same mass to charge ratios, but due to low ion abundance and a low-resolution analyzer, were not taken into consideration for analysis. However, at first glance, this outcome suggests *Cecropia telenitida* has a metabolic capacity of assembling triterpenes resulting in a profile in, which some molecules are isobaric, a positional isomer of epimers.

### 2.2. Classical Molecule-by-Molecule Isolation and NMR Analysis

Given that the main task was identifying the chemical DC16-R fraction profile, classical molecule isolation was selected. Preparative chromatography and several injections were required to obtain a sufficient amount for spectroscopic analysis. A regular chromatogram is shown in Figure 3. All molecules were isolated as described in the extraction process and flash and preparative chromatography section. The 1D 13C-NMR fingerprint for 17.94 min is equal to the previously reported isoyarumic acid [16]. Additionally, the connectivity was fully determined employing homo- and heteronuclear 2D-NMR experiments (see Appendix A).

The NMR-based structure elucidation of the 1D 13C-NMR fingerprint for 23.31 min—despite presenting high homology with the isoyarumic acid 13C-NMR fingerprint—had to be performed de novo due to spectral differences.

The 23.31 min compound presents a doublet signal at 2.74 ppm, the multiplicity-edited 1H–13C HSQC experiment (edited-HSQC) revealed that this proton is attached to an 82.8 ppm signal in 13C-NMR and corresponds to a methine proton. The HMBC spectra indicate coupling with 3.42 ppm proton (67.70 ppm) with a proton whose multiplicity and HMBC coupling reveals CH2 vicinity (0.76–1.75 ppm, edited-HSQC-negative phase). In this order of ideas, both proton signals, 3.42 and 2.74 ppm, were assigned to CH-OH positions 2 and 3, respectively. The double hydroxylation at the positions mentioned is broadly consistent with pentacyclic triterpenes biogenesis.

The presence of a 5.17 ppm proton triplet signal suggests the presence of olefin (edited-HSQC-positive phase). The multiplicity indicates CH2 vicinity, a fact confirmed through HMBC and COSY spectra (1.88 ppm, edited-HSQC-negative phase). The 5.17 ppm signal also showed strong HMBC coupling with 139.10 ppm quaternary carbon (no HSQC correlation observed); based on biosynthetic structural features, the double bond was assigned to the C12-C13 position.

The C13 position (139.10 ppm) shows a correlation with the 2.38 ppm singlet proton (edited-HMBC positive phase) that, in turn, couples with 179.46 ppm carbon. Considering the pentacyclic triterpenes structural features, only C18 or C19 is the closest methine to get a coupling range. Nevertheless, due to the proximity of C17 carboxylic acid, in the aforementioned signal, the chemical shift 2.38 ppm was assigned to the C18 position. The multiplicity of the 2.38 ppm signal also suggests that the C19 position corresponds to quaternary carbon, added to the fact of its coupling with highly displaced carbon (72.13 ppm, no HSQC correlation observed) and 3H singlet (1.09 ppm, edited-HSQC-positive phase). This spectral difference to the 23.31 min compound from isoyarumic acid, due to a shift of the hydroxyl group from position C20 to C19. The adjacent position corresponded to a 0.84 ppm doublet (0.84 ppm, edited-HSQC-positive phase). This molecule was previously reported as tormentic acid [6,17,18] (Figure 4).

From DE16-R, a peak at 27.47 min was also isolated. The 1H-NMR spectrum reveals some structural features that differentiate tormentic from isoyarumic acid despite their double bond position remaining unchanged. Instead of seven signals with 3H integrals (methyl), the 27.47 min sample showed six. The presence of proton signal 3.20–3.35 ppm (69.60 ppm, edited-HSQC-negative phase) coupling with 3.12 and 3.72 ppm proton signals (80.58 and 65.37 ppm, positive phase edited-HSQC) indicates the hydroxylation of one of the dimethyl group at position 4 while revealing hydroxylation at positions 2 and 3 and explaining the absence of the seventh methyl signal in the 1H-NMR spectrum.

The carbonyl signal (179.04 ppm, position 28) and quaternary olefinic carbon (144.01 ppm, position 13) couple with 2.92 ppm signal (43.53 ppm, positive phase edited-HSQC), characteristic of position 18 whose multiplicity, in addition, reveals no vicinity substitution, while COSY coupling permits the assigning of the 2.20 ppm signal to position 19. In mentioning signal position couples to two methyl groups appearing as singlets, which suggest the double substitution at 20 carbon. These structural features imply that the 27.47 min compound corresponds to arjunolic acid [19,20,21].

The peak at 28.57 min showed a characteristic signal for CH2-OH, while methyl group integrals indicate the presence of six instead typically seven substituents. However, unlike arjunolic acid, there are no signals for position 2 hydroxylation. The position of the double bond between C12 and C13, as well as in dimethyl groups, remains unchanged. In accordance with these spectroscopic facts, this sample corresponds to hederagenic acid [22,23]. The lower-area peak at 29.91 retention time was discarded due to human error, and no spectroscopic information was obtained.

### 2.3. Cell Viability in HepG2, C2C12 and 3T3-L1cell Lines

Cell viability was evaluated using the MTT assay after the treatment of corosolic acid and DC16-R fraction for 4 h (C2C12), 48 h (HepG2) or seven days (3T3-L1). These times were used to perform the functional tests. DE16R fraction demonstrated a significant reduction in cell viability at concentrations > 62.5 mg mL^−1^ in HepG2 cells, >125 mg mL^−1^ in C2C12 cells and > 31.3 mg mL^−1^ in 3T3-L1 cells. Corosolic acid is a pentacyclic triterpene not reported in *Cecropia telenitida,* but still presenting the same scaffold of molecules reported in the fraction, and it reduces cell viability at concentrations ≥ 31.3 mg mL^−1^ in HepG2 cells, ≥ 62.2 mg mL^−1^ in C2C12 cells and probably at concentrations ≤ 31.3 mg mL^−1^ in 3T3-L1 cells (Figure 5). Cell viability assays consider that a compound is safe, at values higher than 80% [24].

## 3. Discussion

### 3.1. Biosynthetic Pathway to Pentacyclic Triterpenes and Cecropia telenitida PT Profile

With the aim of describing how *C. telenitida* is able to biosynthesize the isolated PTs in this work, it is necessary to return to the biochemical intermediate squalene, which undergoes enzyme-catalyzed cyclization. The extensive structural diversity of PTs obeys a biogenetic isoprene rule that is able to predict the metabolic outcome. Once oxidosqualene cyclization takes place in the biogenetic pathway, tetracyclic triterpenoids are assembled to produce two different intermediate cations differing in the ring configuration, as shown in Figure 6. Protosteryl cation with a C-B-C configuration that produces sterol compounds, and an all-chair dammaneryl cation configuration that is able to produce a large variety of triterpenoids, and cyclopentane D-ring has a semi-boat configuration as shown in Figure 6. The dammaneryl cation can undergo a ring expansion throughout C16 migration to form baccharenyl cation, which may undergo other ring expansion from C18β migration to form lupyl cation (1). The five-membered E-ring in lupyl cation can either undergo expansion via C18 migration to direct the biosynthetic route to ursanyl cation (4) or C20 migration to form germanicyl cation. The methyl and hydride1,2-shift observed, are generally antiperiplanar, preserving the stereo-position when the shift takes place. The methyl 1,2-shift in germanicyl cation will render the taraxeryl cation (3), and 1,2-hydride shift will render oleanyl cation (2). These cations are quenched by deprotonation or by the addition of a hydroxyl group, and later some cytochrome P450 monooxygenases, CYP716 family (CYP716A265, CYP716A266) catalyze C-28 oxidation to alcohol, to aldehyde and later to carboxylic acid [25,26]. The CYP716C55 enzyme catalyzes the C-2α hydroxylation, resulting in a variety of natural compounds resulting from each cation and responding to the specific stereochemistry. The molecules derived from the cations mention above are shown in Figure 6 as follows: from cation 1: lupeol, lupenone, betulinic acid. Cation 2: oleanolic acid, maslinic acid, b-amyrin arjunolic acid, hederagenic acid, serjanic acid, spergulagenic acid A. Cation 3: taraxasterol; and cation 4:20-hydroxy-ursolic acid, ursolic acid, α-amyrin, corosolic acid, asiatic acid, tormentic acid, isoyarumic acid, yarumic acid, goreishic acid I.

### 3.2. A Brief Overview of the Structure–Activity Relationship among PTs

Some PTs derived from lupyl cation, maintaining some of the oxygenated pattern (at C2, C19, C20 and carboxylic acid at 28), provide the molecules with a specific asset related to modulation of metabolic diseases and type 2 diabetes. For instance, the carboxylic acid at C28 seems to play an important role in interacting with Tyr177 from the 11β-HSD1 enzyme, which is involved in cortisol homeostasis [18], and at the same time, oxygenated C2α position may interact with Thr124 from the same enzyme. All PT compounds discovered from *C. telenitida* fulfill the structural C28 carboxylic acid requisite and, despite low potency, the synergistic effect when they are supplemented at the same time will render systemic and reproducible outcomes [27,28]. The fact of several molecules sharing a closely related scaffold acting at the same time over many targets is part of the benefits offered by this controlled fraction, providing mild potency through a group of molecules acting synergistically while reducing the incidence of side effects. The principal challenge in converting these active substances into phytotherapeutic products or dietary supplements is ensuring their presence, requiring a process of standardizing extracts to guarantee chemical control and reproducible pharmacological effectiveness.

In the same way, those substances, which lack the C28 carboxyl group and have C11 and C12 oxygenated positions like those expressed in Boswellia serrata or in semi-synthetized PTs such as carbenoxolone, they may produce 11β-HSD2 inhibition. Such inhibition is generally detrimental to health due to the accumulation of local cortisol concentration, which causes symptoms of apparent mineralocorticoid excess such as hypertension, fetal developmental defects and lower testosterone levels in males [29,30]. In the case of other triterpene scaffolds dissimilar to derived lupyl cation, for instance, those expressed in ginseng root with a tetracyclic motif, other types of clinical uses have been found [9].

The modification of oxygenated positions using other types of electronegative groups such as cyanides, and functionalization in C17 carboxyl moiety, would render interesting molecules like omaveloxolone (RTA-408) for Friedreich’s Ataxia (mitochondrial myopathy), which is currently undergoing the last stages of clinical trials.

It is evident that humans have eaten pentacyclic triterpenes since ancient times through the consumption of apples, olives, pears, mangos, guava, and many other fruits. These foods contain triterpenes typically ranging from 0.1 to 10 mg g^−1^. In an apple, for instance, the content ranges from 0.28–0.34% of dried peel weight [31], in olive oil close to 40 mg kg^−1^ [1], pears close to 5 mg g^−1^ in dry weight [32]. The most important fact seen in their PT profile is that the most abundant ones are those that conserve the functionalization and oxygenation patterns seen in *C. telenitida*´s PT profile. A person consuming fruits, beverages, extracts, or a dietary supplement will always consume in terms of triterpene content, a group of molecules, which in bulk reach the amounts mentioned above, but never reaching concentrations that would produce toxicity issues [1,33,34,35,36]. With the purpose of analyzing toxicity, the extract DE16-R was compared with a pure isolated substance, corosolic acid, in order to understand how *C. telenitida* has been traditionally used for centuries without known reports of intoxication.

### 3.3. Viability Cell Test and General Considerations

C2C12, HepG2 and 3T3-L1 cell lines are widely used in evaluating the potential activity of new molecules with antidiabetic activity or different metabolic alterations [37]. In this study, results show that the DE16-R fraction exhibits low cytotoxicity at concentrations above 62.5 mg mL^−1^, and corosolic acid at a higher concentration than 31.3 mg mL^−1^ in C2C12 and HepG2 cells. A reduction in cell viability at low concentrations was evident in 3T3-L1 cells, probably because the cells were exposed to the two different compounds for longer. The DE16-R fraction is less toxic than the corosolic acid molecule.

It is important to note that these tests are a preliminary evaluation to assess the viability and safety of the compounds for medication, and more studies must be conducted in order to know the effect of these molecules on healthy cells and the possible control of metabolic alteration associated with obesity and type 2 diabetes mellitus.

## 4. Materials and Methods

### 4.1. Reagents

HPLC grade solvents (dichloromethane, ethyl acetate and acetonitrile LiChrosolv^®^), Mass-Spectrometry grade (acetonitrile LiChromsolv^®^ hypergrade) solvents and ammonium acetate were acquired from Merck (Germany) and used without further purification. The TLC silica gel 60 F_254_ plates from Merck (Germany) were employed to perform preliminary chromatographic fingerprinting.

Chemical and reagents used in cell culture were PBS 1X (Gibco, Carlsbad, CA, USA), low glucose Dulbecco’s modified Eagle’s medium (DMEM) (Sigma-Aldrich, St. Louis, MO, USA), high glucose Dulbecco’s modified Eagle’s medium (DMEM) (Sigma-Aldrich, St. Louis, MO, USA), RPMI 1640 Medium (Sigma-Aldrich, St. Louis, MO, USA), fetal bovine serum (FBS) (Invitrogen, Carlsbad, CA, USA), penicillin–streptomycin (P/S) 10,000 U/Ml (Invitrogen, Carlsbad, CA, USA), glutamine (Gibco, Carlsbad, CA, USA), methyl thiazole tetrazolium (MTT, Amresco, Solon, OH, USA) assay.

### 4.2. Plant Treatment

The *Cecropia telenitida* roots were collected in Rionegro, Antioquia (Colombia) in December 2017 at an altitude of 2254 m.a.s.l. and at the following geodesic location 6°06′36.7′′ N 75°23′21.2′′ W. The plant material was compared and identified against voucher Alzate-Montoya 5189 stored at the Universidad de Antioquia herbarium. The roots were dried at 45 °C for seven days and then subjected to exhaustive milling using a conventional disk sander. The final yield of powdered roots was 1.4 kg.

### 4.3. Extraction Process and Flash and Preparative Chromatography

The powdered roots were extracted using a solvent system composed of dichloromethane (1): ethyl acetate (1) at room temperature (25 °C) and under constant stirring (90 rpm) for 14 days, employing a semi-pilot extraction system designed and manufactured by Process Solutions and Equipment (Bogotá, Colombia). In order to perform the fractionation process, the automated flash chromatography conditions were replicated according to the reported procedure without any modification [16].

DE16R fractions were submitted to purification using a preparative Agilent HPLC 1100 (California, United States). The separation followed at a 210 nm wavelength and was carried out on a Gemini 5 µm C6-Phenyl 110 Å (250 × 10 mm), by means of a gradient elution using water with formic acid 0.1% (A), and acetonitrile also with formic acid 0.1% (B) in a gradient from 0–5 min 40% B, and then 5–55 min increasing up to 95% B.

### 4.4. LC–MS and MALDI-TOF Analysis

LC–MS analysis was performed using an ACQUITY UPLC H-Class system coupled to a single quadrupole mass detector (SQ Detector 2). The atmospheric pressure chemical ionization source (APCI, Waters, Massachusetts, United States) was operated in negative ion mode with a cone voltage of 80 V, a source temperature of 550 °C and a desolvation temperature of 150 °C. The separation was carried out in a column ACQUITY UPLC^®^ BEH C18 (2.1 × 100 mm) with a particle size of 1.7 µm, from Waters (Massachusetts, United States) with gradient elution using ammonium acetate 5 µM pH 9.2 buffer (A) and acetonitrile (B) as follows 40% B → 100% B (0–9 min), 100% B → 100% B (9–11 min), 100% B →40% B (11–15 min), 40% B → 40% B (15–18 min).

MALDI-TOF/TOF analysis was performed using a Bruker Daltonics Ultraflextreme mass spectrometer (Billerica, MA, USA). The instrument is equipped with a 1 kHz Smart Beam Nd:YAG laser (355 nm) operated at 60% of the instrument’s arbitrary power scale (3.92 µJ/pulse). Negative ion mass spectra, from *m*/*z* 200 to 800, were acquired in reflectron mode with a pulsed ion extraction set at 100 ns and an accelerating voltage of 20 kV. Instrument calibration was performed using 1,5-diaminonaphthalene and a mixture of phthalocyanines, purchased from Sigma-Aldrich (St. Louis, MO, USA), covering the entire working mass range. Each reported analysis corresponds to the sum of 5000 mass spectra. Data analysis was performed using the Flex Analysis software version 3.4 (Bruker Daltonics, Billerica, MA, USA).

### 4.5. Nuclear Magnetic Resonance Experiments

The Nuclear Magnetic Resonance experiments were performed on a Bruker Ascend III HD 600 MHz spectrometer (BioSpin GmbH, Rheinstetten, Germany) equipped with a 5 mm cryoprobe—TCI using DMSO-*d*_6_ as a solvent.

### 4.6. Cell Cultures and Cell Viability Test

The C2C12 (ATCCCRL-1772TM) mouse muscle cells, 3T3-L1 (CL-173™) mouse fibroblast cells and HepG2 (ATCC HB-8065TM) human liver carcinoma cells were purchased from ATCC (Manassas, VA, USA). Cells were cultured and maintained at 37 °C and 5% CO_2_ in DMEM culture medium with 10% fetal bovine serum (FBS), 2 mM glutamine, penicillin and 1% streptomycin (Sigma). When the C2C12 cells reached a confluence between 80 and 90%, they were differentiated into myotubes, using low glucose DMEM (5.5 mM) medium supplemented with 5% horse serum (HS) [38].

3T3-L1 cells were cultured in DMEM with 10% FBS, 1% P/S and 25 mM glucose (Growth Medium 1–GM1). Differentiation was induced two days post-confluence by adding GM1 medium containing 0.5 mM IBMX, 0.25 μM dexamethasone, 2 μM Rosiglitazone and 1 μg/mL insulin. After two days of incubation, the medium was changed to a GM1 medium containing 1 μg/mL insulin. Two days later, the medium was replaced by GM1 and incubated for another seven days with the different treatments. Corosolic acid, an antidiabetic compound, was used as the control. In addition, corsolic acid has the same PT scaffold present in *Cecropia telenitida*’s triterpenes, but it has not been reported in the plant.

To evaluate cell viability, 3T3-L1, HepG2 and C2C12 cells were seeded on 96 multi-well plates at 2.5 × 104 cells/well and cultured for 24 h. The cells were washed with DMEM once and then incubated with different concentrations of corosolic acid (CA) and DE16 fraction (250, 125, 62.5, 31.25 μg/mL) in DMEM for 4h (C2C12), 48 h (HepG2) and seven days (3T3-L1). Subsequently, the medium was removed, cells were washed once, and MTT 5 mg/mL in DMEM were added to each well and incubated for 4 h. The MTT medium was removed, and 200 μL of DMSO was added to dissolve the formazan formed. The optical densities (OD) were measured using a Varioskan Flash spectrophotometer (Thermo, Waltham, MA, USA) at 570 nm [39].

## 5. Conclusions

In conclusion, we have shown that *Cecropia telenitida* can assemble isobaric, positional isomers or epimeric pentacyclic triterpenes. Ursane or oleanane scaffolds with the same oxygenation pattern are always shared by the PTs in *C. telenitida*, as demonstrated by its biosynthetic pathway. As *Cecropia telenitida* has been used in traditional medicine by local communities, we believe this information may be useful to explain how a group of molecules sharing a closely related scaffold can provide medicinal efficacy.

As seen in the viability assay, a fraction of PTs compared to a pure one evaluated under similar conditions and at similar concentrations decreases potential toxicity. The regulatory landscape worldwide brings up a relevant discussion regarding identifying toxic constituents of botanical dietary supplements, and the main concern is mainly focused on demonstrating safety. Several decades of traditional human use with chemical control and in vitro toxicity over bioactive markers comprises the first steps of long but promising research with this plant. We are convinced that phytomedicine is a reliable and safe type of medication, but always by means of a chemically controlled, multicomponent nutritional supplement or phytotherapeutic product.

## Figures and Tables

**Figure 1 molecules-26-01064-f001:**
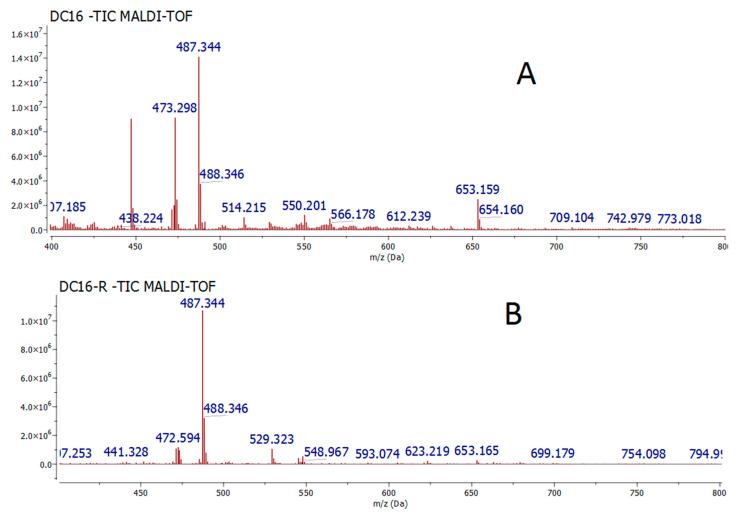
(**A**) Mass spectra of originally obtained leader fraction (DE16). (**B**) Mass spectra of fractions pooled (DE16-R). Ionization was carried out by MALDI (−) using DMAN as a matrix.

**Figure 2 molecules-26-01064-f002:**
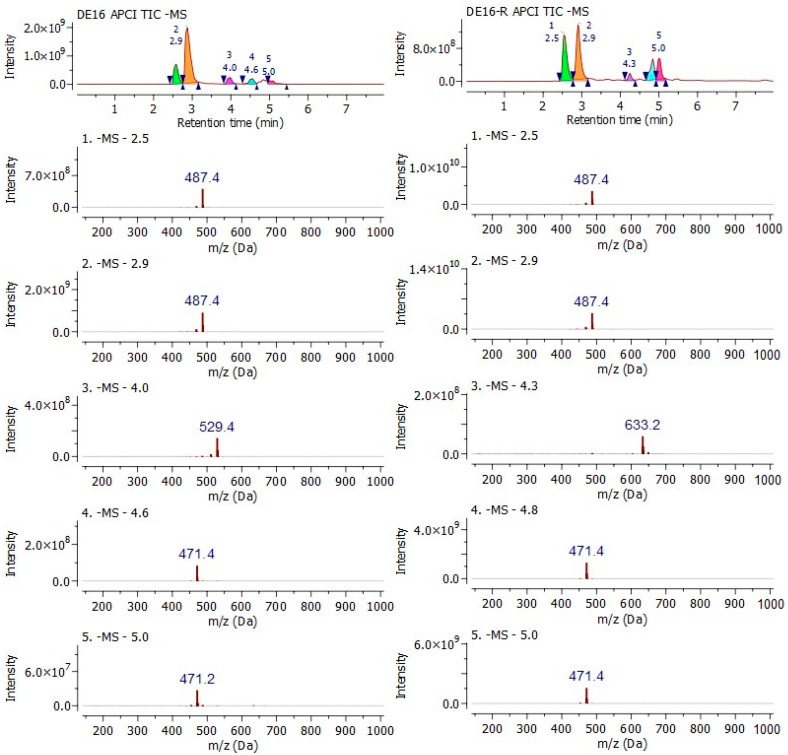
UPLC-APCI-MS negative mode chromatograms for DE16 and DE16-R fractions. The total ion chromatogram (TIC) spectra were obtained after peak integration and subtracting the solvent noise. Every spectrum is extracted, with each peak designated a consecutive number from 1 to 5.

**Figure 3 molecules-26-01064-f003:**
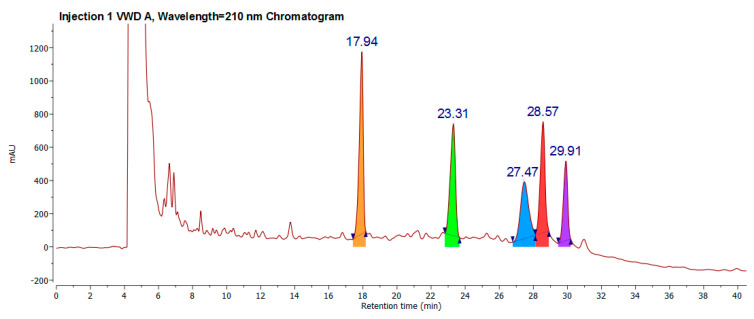
A preparative chromatogram obtained by HPLC-DAD from DE16R. The 17.94 retention time molecule was identified as isoyarumic acid, 23.31 min was identified as tormentic acid, 27.47 min was identified as arjunolic acid, and 28.27 min was identified as hederagenic acid.

**Figure 4 molecules-26-01064-f004:**
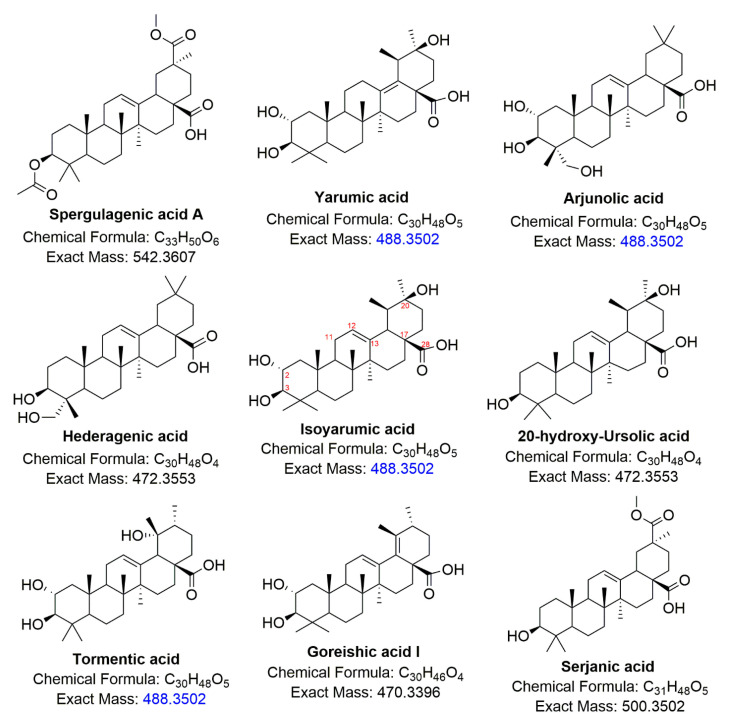
Presentation of pentacyclic triterpene (PT) structures found in *Cecropia telenitida*. Isoyarumic acid, tormentic acid, arjunolic acid, and hederagenic acid are molecules isolated and identified as an outcome of this work. The exact masses highlighted in blue show isobaric triterpenes.

**Figure 5 molecules-26-01064-f005:**
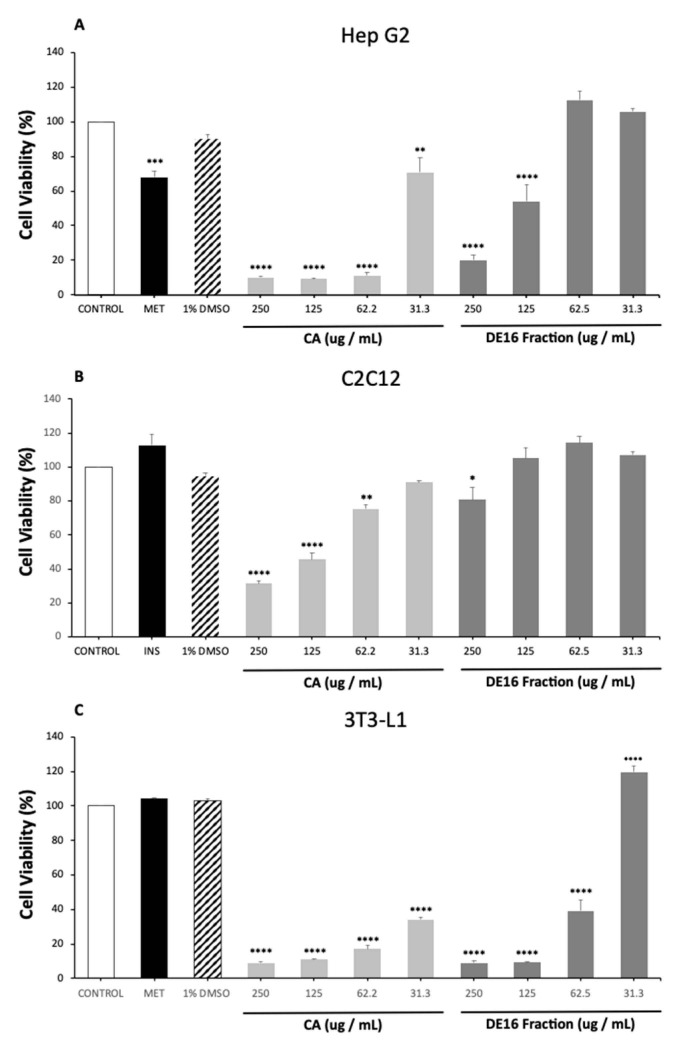
Cytotoxicity of corosolic acid (CA) and DE16-R fraction on cell lines 3T3-L1, HepG2 and C2C12. The cells were seeded in 96-well plates and treated for 48H (HepG2) (**A**), 4 h (C2C12) (**B**) and 7 days (3T3-L1) (**C**). The cytotoxicity was evaluated by methyl thiazole tetrazolium (MTT) assay. Values are expressed as means ± SEM. ANOVA with post hoc Dunnett’s for multiple comparisons was performed. ****: *p* < 0.0001; ***: *p* < 0.001; **: *p* < 0.01; *: *p* < 0.05. *n* = 3.

**Figure 6 molecules-26-01064-f006:**
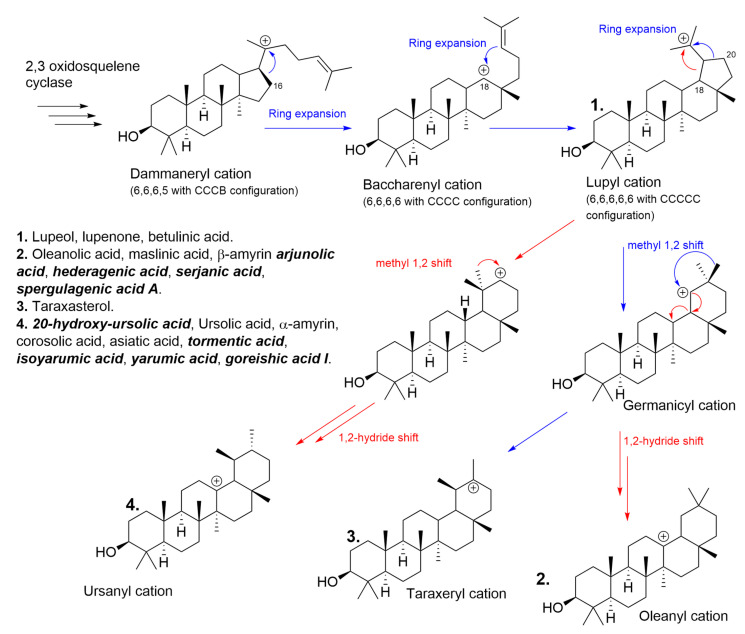
Biosynthetic pathways to producing PTs. The underlined names refer to molecules isolated and identified for the first time in this plant. The nine molecule names bolded and italicized mark the nine PTs reported.

## Data Availability

Not applicable.

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
