# Peer review of "Pentacyclic Triterpene Profile and Its Biosynthetic Pathway in Cecropia telenitida as a Prospective Dietary Supplement"

_molecules, 2021, doi:10.3390/molecules26041064_

Round 1

Reviewer 1 Report

The author explains the medicinal benefits of C.telenitida in type 2 diabetes. The use of traditional medicine will help the local community. Although the subject is interesting, especially due to the use of phytomedicine supplementation on human health. Some important concerns need to be addressed before the manuscript is ready for publication.

  • An in vivo/in vitro study needs to confirm its role in type 2 diabetes.
  • Give details of pharmacokinetics and pharmacodynamics of pentacyclic triterpene.
  • Do pentacyclic triterpenes isolated from telenitida have a similar metabolism to others?

Author Response

Reviewer #1:

Statement #1: The author explains the medicinal benefits of C. telenitida in type 2 diabetes. The use of traditional medicine will help the local community. Although the subject is interesting, especially due to the use of phytomedicine supplementation on human health. Some important concerns need to be addressed before the manuscript is ready for publication.

Appreciative statement, no answer required.

Statement #2: An in vivo/in vitro study needs to confirm its role in type 2 diabetes.

The present work is a continuous research of our previously published paper titled: “Pentacyclic Triterpenes from Cecropia telenitida Can Function as Inhibitors of 11β-Hydroxysteroid Dehydrogenase Type 1”. In that work we assessed the in vitro inhibitory activity of DE16 fraction against 11β-HSD1, the enzyme responsible for cortisone activation to cortisol. The role of cortisol in type 2 diabetes was widely discussed and reported.

Joseph, J. J., & Golden, S. H. (2017). Cortisol dysregulation: the bidirectional link between stress, depression, and type 2 diabetes mellitus. Annals of the New York Academy of Sciences, 1391(1), 20.

Basu, A., Yadav, Y., Carter, R. E., & Basu, R. (2020). Novel Insights into Effects of Cortisol and Glucagon on Nocturnal Glucose Production in Type 2 Diabetes. The Journal of Clinical Endocrinology & Metabolism, 105(7), e2378-e2388.

Dammann, C., Stapelfeld, C., & Maser, E. (2019). Expression and activity of the cortisol-activating enzyme 11β-hydroxysteroid dehydrogenase type 1 is tissue and species-specific. Chemico-biological interactions, 303, 57-61.

Steptoe, A., Hackett, R. A., Lazzarino, A. I., Bostock, S., La Marca, R., Carvalho, L. A., & Hamer, M. (2014). Disruption of multisystem responses to stress in type 2 diabetes: investigating the dynamics of allostatic load. Proceedings of the National Academy of Sciences, 111(44), 15693-15698.

In this order of ideas, our fraction was previously tested as 11β-HSD1 inhibitor and the present works pursue two mains goals: i) Serve as an evidence of fraction obtainment reproducibility in terms of composition ii) Increase our understanding about the chemical space associated to DE16 fraction and iii) To sssess the toxicity of this fractions over different cellular lines just as confirmatory data since these family compounds has toxicology endorsed by use and was evaluated in vivo in our previous work.

Gutiérrez, G., Giraldo-Dávila, D., Combariza, M. Y., Holzgrabe, U., Tabares-Guevara, J. H., Ramírez-Pineda, J. R., ... & Balcazar, N. (2020). Serjanic Acid Improves Immunometabolic Markers in a Diet-Induced Obesity Mouse Model. Molecules, 25(7), 1486

Statement #3: Give details of pharmacokinetics and pharmacodynamics of pentacyclic triterpene.

We have discussed this statement and we have decided no to include it, based on three main reasons. The first one is because the most of information describing pharmacokinetics were carried out with close related molecules like ursolic, oleanolic, and maslinic acids, and unfortunately not with the identified molecules. The second reason is because there is a published review dealing with this specific topic (10.3390/molecules22030400). And the last one is because regulatory agencies do not require bioavailability information to approve to reach the market to dietary supplements.

Despite Dietary supplement will expected to be safe, for new dietary ingredients (like we are dealing with -herbs or botanical) the most important information is related with the identification of hazardous issues and the identification of toxicity. Currently we are running research projects dealing with pharmacokinetics and a complete safety profile on biomodels, and we hope to be publishing in a frame of two years.

Statement #4: Do pentacyclic triterpenes isolated from telenitida have a similar metabolism to others?

As was mentioned above, this is a completely different topic and the only way to answer it, is by funding a new research project. To answer to this would be speculative

Reviewer 2 Report

Dear Authors,

This manuscript is a continuous work of Authors, which has published in Molecules 2018, where a followed up work has led to the isolation of four known triterpenes from Cecropia telenitida. In addition, the cytotoxicity was evaluated which contributed for future research regarding these terpenes and species.

1) The Supporting Data was not provided but it was mentioned in the manuscript at lines 143.

2) At lines 144-172, discussing the NMR features that led to structure determination of tormentic acid for HPLC peak at 23.31 min. I felt that these sentences are too long for characterizing a known compound, I suggest Authors might consider to shorten it. 

3) At lines 177-191, the above suggestion is applying to here, again, I think Authors might consider to shorten the result part for characterizing HPLC peak at 27.47 min as arjunolic acid (known). I believed that the NMR profile of isoyarumic, tormentic and arjunolic acids are distinct to each other, although their structures are relatively identical, therefore by comparison of 1H and 13C NMR should be enough to distinguish and identify these compounds.

4) At 2.3 section, mg mL-1, where -1 should be superscripted, at lines such as 203. A positive control is not require to perform the assay?

5) HPLC peak at 29.91 min was not mention why it was not identify.

6) At 3.1 section, I think Authors might refrain from using "Biosynthetic pathways to producing PTs in the roots of Cecropia telenitida.", instead please replace with purposed biosynthetic pathway, if the biosynthetic pathway of these triterpenes were not reported to date. If it was reported, please cite the original paper as reference.

Author Response

Reviewer #2:

Dear Authors,

Statement #1: This manuscript is a continuous work of Authors, which has published in Molecules 2018, where a followed up work has led to the isolation of four known triterpenes from Cecropia telenitida. In addition, the cytotoxicity was evaluated which contributed for future research regarding these terpenes and species.

Appreciative statement, no answer required.

Statement #2: The Supporting Data was not provided but it was mentioned in the manuscript at lines 143.

We provided the material in an extemporaneous way. We offer excuses and kindly invite you to check again the platform where you will find the supplementary information.

Statement #3: At lines 144-172, discussing the NMR features that led to structure determination of tormentic acid for HPLC peak at 23.31 min. I felt that these sentences are too long for characterizing a known compound, I suggest Authors might consider to shorten it. At lines 177-191, the above suggestion is applying to here, again, I think Authors might consider to shorten the result part for characterizing HPLC peak at 27.47 min as arjunolic acid (known). I believed that the NMR profile of isoyarumic, tormentic and arjunolic acids are distinct to each other, although their structures are relatively identical, therefore by comparison of 1H and 13C NMR should be enough to distinguish and identify these compounds.

We are agreed on your appreciation that there is sufficient NMR profile difference between the isolated compounds despite the high structural homology. Nevertheless, precisely due to that appreciation we think that is important to describe the spectral features in a detailed way since the exact chemical shift depends on solvent and equipment. We also provided highly detailed spectral evidence on supplementary information since from our perspective it is important for transparency purposes.

Statement #4 At 2.3 section, mg mL-1, where -1 should be superscripted, at lines such as 203. A positive control is not require to perform the assay?

The superscript for every single -1 were corrected in whole manuscript.

On the other hand, the positive control in the cytotoxicity is not required since negative control is showing that the vehicle treatment is not cytotoxic and cell growth methodology is working properly and cell viability showed and dosage-response trend, these allow us to conclude that the assay is capable to measure the toxicity.

Statement #6 HPLC peak at 29.91 min was not mention why it was not identify.

The peak was discarded due to not enough purity and abundance during the isolation, it is stated on lines 199-200.

Statement #7 At 3.1 section, I think Authors might refrain from using "Biosynthetic pathways to producing PTs in the roots of Cecropia telenitida.", instead please replace with purposed biosynthetic pathway, if the biosynthetic pathway of these triterpenes were not reported to date. If it was reported, please cite the original paper as reference.

We agreed the modified the figure 6 caption as follow: “Figure 6. Biosynthetic pathways to producing PTs. The underlined names refer to molecules isolated and identified for the first time in this plant. The nine molecule names bolded and italicized mark the nine PTs reported.”

Reviewer 3 Report

The manuscript entitled: Pentacyclic triterpene profile and its biosynthetic pathway in Cecropia telenitida as a prospective type 2 diabetes dietary supplement, is an original and very interesting work.   The authors present a very important phytochemical analysis. The chemical analyzes were carried out correctly, obtaining a very interesting chemical profile.   However, the association with biological systems and its possible effect is not clear. The authors evaluated the effect of a compound and a fraction on cell viability in three cell lines, but the intent of the assay is unclear. Cytotoxicity and cytotoxic effect is not a characteristic activity of this type of compound. Toxicological approaches must focus on an animal model. I believe that the authors should complement with more studies, tests directed to the biological effect of these compounds, using animal or cellular models. This will allow to give solid sustenance to the potential phytomedicine that can be developed. The authors could evaluate adipogenesis, glucose incorporation, insulin secretion, agonist activity to PPARs.

The authors could evaluate the effect of the compound or fraction in an animal model of obesity, diabetes, or metabolic syndrome. These approaches could complement the advance in the investigation of these compounds

Author Response

Reviewer #3

Statement #1: The manuscript entitled: Pentacyclic triterpene profile and its biosynthetic pathway in Cecropia telenitida as a prospective type 2 diabetes dietary supplement, is an original and very interesting work.   The authors present a very important phytochemical analysis. The chemical analyzes were carried out correctly, obtaining a very interesting chemical profile.   

No answer required.

Statement #2: However, the association with biological systems and its possible effect is not clear. The authors evaluated the effect of a compound and a fraction on cell viability in three cell lines, but the intent of the assay is unclear. Cytotoxicity and cytotoxic effect is not a characteristic activity of this type of compound.

The present work is a continuous research of our previously published paper titled: “Pentacyclic Triterpenes from Cecropia telenitida Can Function as Inhibitors of 11β-Hydroxysteroid Dehydrogenase Type 1”. In that work we assessed the in vitro inhibitory activity of DE16 fraction against 11β-HSD1, the enzyme responsible for cortisone activation to cortisol. The role of cortisol in type 2 diabetes was widely discussed and reported.

Joseph, J. J., & Golden, S. H. (2017). Cortisol dysregulation: the bidirectional link between stress, depression, and type 2 diabetes mellitus. Annals of the New York Academy of Sciences, 1391(1), 20.

Basu, A., Yadav, Y., Carter, R. E., & Basu, R. (2020). Novel Insights into Effects of Cortisol and Glucagon on Nocturnal Glucose Production in Type 2 Diabetes. The Journal of Clinical Endocrinology & Metabolism, 105(7), e2378-e2388.

Dammann, C., Stapelfeld, C., & Maser, E. (2019). Expression and activity of the cortisol-activating enzyme 11β-hydroxysteroid dehydrogenase type 1 is tissue and species-specific. Chemico-biological interactions, 303, 57-61.

Steptoe, A., Hackett, R. A., Lazzarino, A. I., Bostock, S., La Marca, R., Carvalho, L. A., & Hamer, M. (2014). Disruption of multisystem responses to stress in type 2 diabetes: investigating the dynamics of allostatic load. Proceedings of the National Academy of Sciences, 111(44), 15693-15698.

In this order of ideas, our fraction was previously tested as 11β-HSD1 inhibitor and the present works pursue two mains goals: i) Serve as an evidence of fraction obtention reproducibility in terms of composition ii) Increase our understanding about the chemical space associated to DE16 fraction and iii) Assess the toxicity of this fractions over different cellular lines just as confirmatory data since these family compounds has toxicology endorsed by use and was evaluated in vivo in our previous work.

Gutiérrez, G., Giraldo-Dávila, D., Combariza, M. Y., Holzgrabe, U., Tabares-Guevara, J. H., Ramírez-Pineda, J. R., ... & Balcazar, N. (2020). Serjanic Acid Improves Immunometabolic Markers in a Diet-Induced Obesity Mouse Model. Molecules, 25(7), 1486

For the above, the cytotoxicity evaluation was not carried out with the aim to demonstrate that potential biological activity but with the aim of validate the absence of toxicity previously demonstrated using Serjanic Acid, other pentacyclic triterpene isolated from C. telenitida.

Statement #3: Toxicological approaches must focus on an animal model. I believe that the authors should complement with more studies, tests directed to the biological effect of these compounds, using animal or cellular models. This will allow to give solid sustenance to the potential phytomedicine that can be developed. The authors could evaluate adipogenesis, glucose incorporation, insulin secretion, agonist activity to PPARs. The authors could evaluate the effect of the compound or fraction in an animal model of obesity, diabetes, or metabolic syndrome. These approaches could complement the advance in the investigation of these compounds.

We have identified different pharmacodynamic interactions of C. telenitida pentacyclic triterpenes that give them the proposed therapeutic activity:

  1. The 11β-HSD1 inhibition that, as mentioned in statement #2 answer, is capable to modulate different biochemical pathways involved in DMT2 pathogenesis and progression. In mention inhibitory activity was assessed on our previous work.

Mosquera, C., Panay, A. J., & Montoya, G. (2018). Pentacyclic triterpenes from Cecropia telenitida can function as inhibitors of 11β-hydroxysteroid dehydrogenase type 1. Molecules, 23(6), 1444.

  1. Different inflammatory and lipid metabolism pathways are also modulated by TPs isolated from telenitida, it was assessed through metabolic syndrome murine model using Serjanic Acid as model compound.

Gutiérrez, G., Giraldo-Dávila, D., Combariza, M. Y., Holzgrabe, U., Tabares-Guevara, J. H., Ramírez-Pineda, J. R., ... & Balcazar, N. (2020). Serjanic Acid Improves Immunometabolic Markers in a Diet-Induced Obesity Mouse Model. Molecules, 25(7), 1486

Please note that C. telenitida is not a commercial crop, we depend on natural individuals and these are not killed during the roots sampling as a good practice in sustainable natural products research, the available quantity of roots is extremely limited at the time that the yield is below 1%. For this reason, the Colombia´s Ministry of Science is financing us to develop the in vitro production of C. telenitida root cells in order to give us the capacity of DE16 production necessary to support further animal model using the fraction and not a model compound

Reviewer 4 Report

There are some minor errors within the paper e.g.

Abstract

  • “Profile of DE16-R”. Please clarify what DE16-R is.
  • “Through the biosynthetic pathway connected to the isolated PTs in C.telenitida, it offers a specific medicinal benefit related to the modulation of T2D”. Please consider revising this sentence as it does not make sense.

Please read through the paper carefully and adjust minor issues.

One major revision would be to correct figure 6 so that the synthetic scheme reads more logically. I think the position of the carbocation in Oleanyl (Fig 6) is incorrect if we are assuming a 1,2-hydride shift based on the structure of the Germanycil cation provided.

Author Response

Manuscript Review Round 1: Pentacyclic triterpene profile and its biosynthetic pathway in Cecropia telenitida as a prospective type 2 diabetes dietary supplement

Reviewer #4

Statement #1 In the abstract section: “Profile of DE16-R”. Please clarify what DE16-R is.

The lines # 87 and 88 were modified to clarify the DE16-R meaning as follows: “To this end, we studied the compositional PTs profile of the leader fraction obtained from other individual (DE16-R) through MALDI-TOF(…)” We did not included on abstract section for shortening purposes.

Statement #2 In the abstract section: “Through the biosynthetic pathway connected to the isolated PTs in C. telenitida, it offers a specific medicinal benefit related to the modulation of T2D”. Please consider revising this sentence as it does not make sense.

Statement #3 Please read through the paper carefully and adjust minor issues.

Not require to be answered

Statement #4 One major revision would be to correct figure 6 so that the synthetic scheme reads more logically. I think the position of the carbocation in Oleanyl (Fig 6) is incorrect if we are assuming a 1,2-hydride shift based on the structure of the Germanycil cation provided.

The figure 6 was modified to correct the mechanism and depurating captions and other information not relevant for the purpose of figure 6 in our manuscript.

Round 2

Reviewer 3 Report

The title of the manuscript describes the biosynthetic pathway of Pentacyclic triterpen compounds identified in Cecropia telenitida. The compounds are proposed as potential compounds for the dietary supplement of patients with type 2 diabetes. However, the authors do not show clear evidence regarding their potential use. The authors only show the potential cytotoxic effect of these compounds on three cell lines. these data do not support the nomination of such compounds for potential treatment or use in T2D. The authors comment that they have previous reports that demonstrate its potential use. The study on these compounds has been done for several years. However, it is important to note that this evidence is not included in this manuscript. therefore, they should not keep the current title, because it does not reflect the content of the manuscript. I suggest modifying the title of the manuscript to reflect the evidence that the authors report in this manuscript, which is only limited to the identification of compounds and the cytotoxic evaluation in three cell lines. 

Author Response

Comments and suggestions

  1. The title of the manuscript describes the biosynthetic pathway of Pentacyclic triterpen compounds identified in Cecropia telenitida. The compounds are proposed as potential compounds for the dietary supplement of patients with type 2 diabetes. However, the authors do not show clear evidence regarding their potential use. The authors only show the potential cytotoxic effect of these compounds on three cell lines. these data do not support the nomination of such compounds for potential treatment or use in T2D. The authors comment that they have previous reports that demonstrate its potential use. The study on these compounds has been done for several years. However, it is important to note that this evidence is not included in this manuscript. therefore, they should not keep the current title, because it does not reflect the content of the manuscript. I suggest modifying the title of the manuscript to reflect the evidence that the authors report in this manuscript, which is only limited to the identification of compounds and the cytotoxic evaluation in three cell lines.

Response:

A rich discussion was carried out with authors of the manuscript due the huge implications to change the title. We agreed to eliminate “T2D” from the title, but still having it in the manuscript key words. The title has changed from:

Pentacyclic triterpene profile and its biosynthetic pathway in Cecropia telenitida as a prospective type 2 diabetes dietary supplement

to

Pentacyclic triterpene profile and its biosynthetic pathway in Cecropia telenitida as a prospective dietary supplement.

  1. Additional modifications were made in the manuscript with the track changes on. Few slight rephrasing in the abstract (lines 27-28)

Figure 6 is modified to amend “Germanicyl cation”

In conclusion section a short paragraph was included

No citation was added to support some points of view.
